

# Investigating the running abilities of *Tyrannosaurus rex* using stress-constrained multibody dynamic analysis

William I. Sellers[1], Stuart B. Pond[2], Charlotte A. Brassey[3], Philip L. Manning[1,4] and Karl T. Bates[5]

[1] School of Earth and Environmental Sciences, University of Manchester, Manchester, United Kingdom
[2] Ocean and Earth Science, National Oceanography Centre, University of Southampton, Southampton, United Kingdom
[3] School of Science and the Environment, The Manchester Metropolitan University, Manchester, United Kingdom
[4] Department of Geology and Environmental Geosciences, College of Charleston, Charleston, United States of America
[5] Department of Musculoskeletal Biology, Institute of Aging and Chronic Disease, University of Liverpool, Liverpool, United Kingdom

Corresponding author
William I. Sellers,
William.Sellers@manchester.ac.uk

## ABSTRACT

The running ability of *Tyrannosaurus rex* has been intensively studied due to its relevance to interpretations of feeding behaviour and the biomechanics of scaling in giant predatory dinosaurs. Different studies using differing methodologies have produced a very wide range of top speed estimates and there is therefore a need to develop techniques that can improve these predictions. Here we present a new approach that combines two separate biomechanical techniques (multibody dynamic analysis and skeletal stress analysis) to demonstrate that true running gaits would probably lead to unacceptably high skeletal loads in *T. rex*. Combining these two approaches reduces the high-level of uncertainty in previous predictions associated with unknown soft tissue parameters in dinosaurs, and demonstrates that the relatively long limb segments of *T. rex*—long argued to indicate competent running ability—would actually have mechanically limited this species to walking gaits. Being limited to walking speeds contradicts arguments of high-speed pursuit predation for the largest bipedal dinosaurs like *T. rex*, and demonstrates the power of multiphysics approaches for locomotor reconstructions of extinct animals.

## INTRODUCTION

*Tyrannosaurus rex* is one of the largest bipedal animals to have ever evolved and as such it represents a useful model organism for understanding morpho-functional adaptations and constraints at multi-tonne body sizes (*Brusatte et al., 2010*). The running ability of *T. rex* and other similarly giant dinosaurs has been intensely debated in the literature (*Bakker, 1986*; *Hutchinson & Garcia, 2002*; *Paul, 1998*; *Paul, 2008*; *Sellers & Manning, 2007*)

and features prominently in reconstructions of the lifestyles and carnivorous behaviours of large theropod dinosaurs (*Bakker, 1986*; *Carbone, Turvey & Bielby, 2011*; *Farlow, 1994*; *Holtz Jr, 2008*; *Paul, 1998*; *Paul, 2008*; *Ruxton & Houston, 2003*). However, despite a century of research since *Osborn*'s *(1916)* work on tyrannosaur limb anatomy there remains no consensus on the most accurate maximum speeds for *T. rex*, or indeed whether or not its gigantic body size prohibited running completely.

Some qualitative anatomical studies (*Bakker, 1986*; *Paul, 1998*; *Paul, 2008*), including some employing a degree of quantitative biomechanical methods (*Paul, 1998*), have proposed very fast running speeds (up to 20 ms$^{-1}$) and an overall high degree of athleticism for large theropods like *T. rex*. These studies cite the long and gracile limbs of *T. rex* as a key adaptive feature indicative of high relative (*Christiansen, 1998*) and absolute speeds (*Bakker, 1986*; *Paul, 1998*; *Paul, 2008*), along with possession of large tail-based hip extensor musculature (*Persons & Currie, 2011*). In contrast, more direct and quantitative biomechanical approaches have favoured intermediate (*Farlow, Smith & Robinson, 1995*; *Sellers & Manning, 2007*) or much slower speeds for *T. rex*, with the latter including within their predictive range an inability to reach true running gaits (*Gatesy, Baker & Hutchinson, 2009*; *Hutchinson, 2004b*; *Hutchinson & Garcia, 2002*). Biomechanical approaches emphasize the well-known scaling principles (*Biewener, 1989*; *Biewener, 1990*) that animals of larger body mass have more restricted locomotor performance because muscle mass scales isometrically, but muscle force, relative speed of contraction and power scale with negative allometry (*Alexander, 1977*; *Alexander & Jayes, 1983*; *Marx, Olsson & Larsson, 2006*; *Medler, 2002*).

Biomechanical models inherently incorporate anatomical characters (e.g., limb proportions) on which more traditional qualitative assessments are based, but also require quantitative definitions for soft tissue parameters associated with mass distribution and muscle properties. These soft tissue parameters are almost never preserved in dinosaur fossils and therefore need to be estimated indirectly. Typically, minimum and maximum bounds are placed on such parameters based on data from living animals (*Hutchinson, 2004a*; *Hutchinson, 2004b*; *Hutchinson & Garcia, 2002*) and/or additional computer models (*Bates, Benson & Falkingham, 2012*; *Bates et al., 2010*; *Hutchinson et al., 2005*; *Sellers et al., 2013*). However, these approaches yield very broad ranges for soft tissue parameters in dinosaurs which translates directly into imprecise values for performance estimates like running speed (*Bates et al., 2010*). Thus, while biomechanical approaches are more explicit and direct by their inclusion of all major anatomical and physiological factors determining running ability, their utility within palaeontology in general has been severely restricted by high levels of uncertainty associated with soft tissues. Consequently, estimates for *T. rex* running speed from biomechanical models range from 5 to 15 m/s (*Gatesy, Baker & Hutchinson, 2009*; *Hutchinson, 2004b*; *Hutchinson & Garcia, 2002*; *Sellers & Manning, 2007*).

One solution is to find information in the preserved skeletal morphology that can be used to reduce the predictive dependence of biomechanical models on soft tissue. It has recently been suggested that bone loading can be used to improve the locomotor reconstruction of fossil vertebrates by excluding gaits that lead to overly high skeletal loads (*Sellers et*

*al., 2009*). It is highly likely that in many cases the skeletons of cursorial vertebrates are optimised for locomotor performance such that the peak locomotor stresses are 25–50% of their failure strength, indicating a safety factor of between two and four (*Biewener, 1990*). There are notable exceptions where long bones are considerably stronger than required (*Brassey et al., 2013a*) but in general this trade-off between body mass and load bearing ability appears to be a widespread anatomical adaptation that is found in invertebrates as well as vertebrates (*Parle, Larmon & Taylor, 2016*). Our previous simulations of theropod bipedal running (*Sellers & Manning, 2007*) did not directly consider the skeletal loading but these simulations do calculate the joint reaction forces and these can be used directly to estimate the bone loading using the beam mechanic methodology described (*Brassey et al., 2013c*). Results for *Struthio camelus* and *T. rex* are shown in Fig. 1 and whilst the values for *Struthio* are easily within those predicted by safety factor analysis, those for *T. rex* would likely exceed the yield strength for bone (approx. 200 MPa (*Biewener, 1990*)). This agrees with more sophisticated results based on ostrich finite element analysis suggesting high safety factors for this species that might relate to non-locomotor activities (*Gilbert, Snively & Cotton, 2016*).

However there are two main problems with approaches based on joint reaction forces. Firstly to accurately calculate the loads sustained *in vivo* during high speed locomotion requires the integration of a large number of different force components from soft tissues, joints, substrate interactions and body segment accelerations, and not simply the joint reaction forces. Estimates can be made using quasi static approaches (e.g., *Gilbert, Snively & Cotton, 2016*) but virtual robotic approaches such as multibody dynamics (MBDA) allow calculation of the complete dynamic loading environment which can then be used to estimate bone loading through beam mechanics (e.g., *Sellers et al., 2009*) or other simulation approaches like finite element analysis (FEA) (e.g., *Curtis et al., 2008*; *Snively & Russell, 2002a*). Secondly it is important that the optimsiation goal includes all the important conditions directly. It is entirely possible that our previous high values for *T. rex* skeletal stress are because the genetic algorithm was only looking for the fastest possible gait. There may be gaits that are only slightly less fast but have a much lower skeletal stress and these may be overlooked if skeletal stress is not considered within the machine learning process. Herein we demonstrate the predictive power of using an integrated approach in palaeontology by combining MBDA, machine learning algorithms and stress analysis to reconstruct maximum locomotor speed in *T. rex*. Machine learning algorithms are used to generate the muscle activation patterns that simultaneously produce the maximum locomotor speed of a MBDA model of *T. rex* whilst maintaining defined skeletal safety factors. Combining the two simulation systems (so-called multiphysics simulation) means that only solutions that satisfy all the criteria are allowable and this should therefore narrow the predicted range of our performance estimates.

## MATERIALS AND METHODS

MBDA approaches to locomotor reconstruction require a linked segment model of the animal to be built based on its skeletal morphology and inferred myology (Fig. 2). The
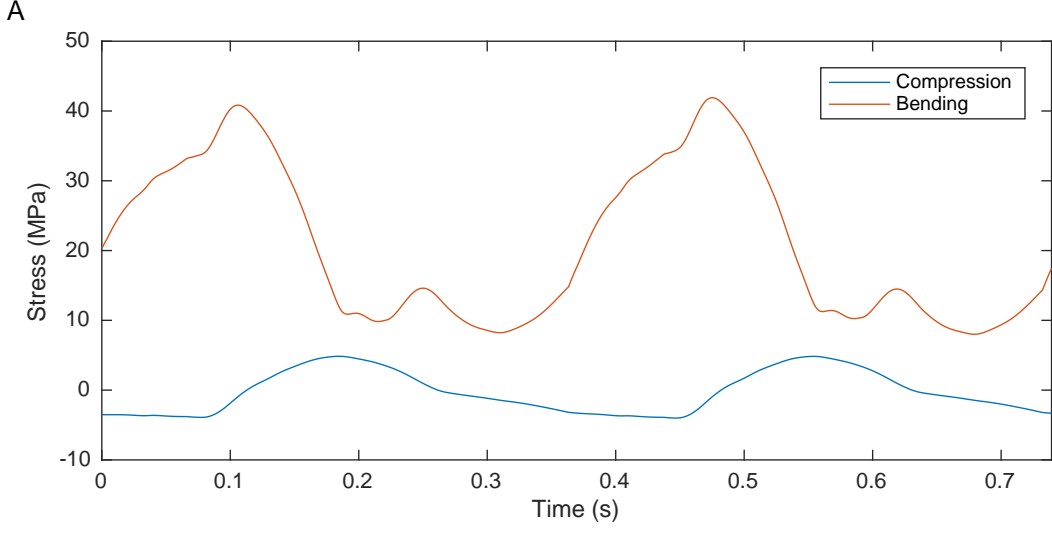

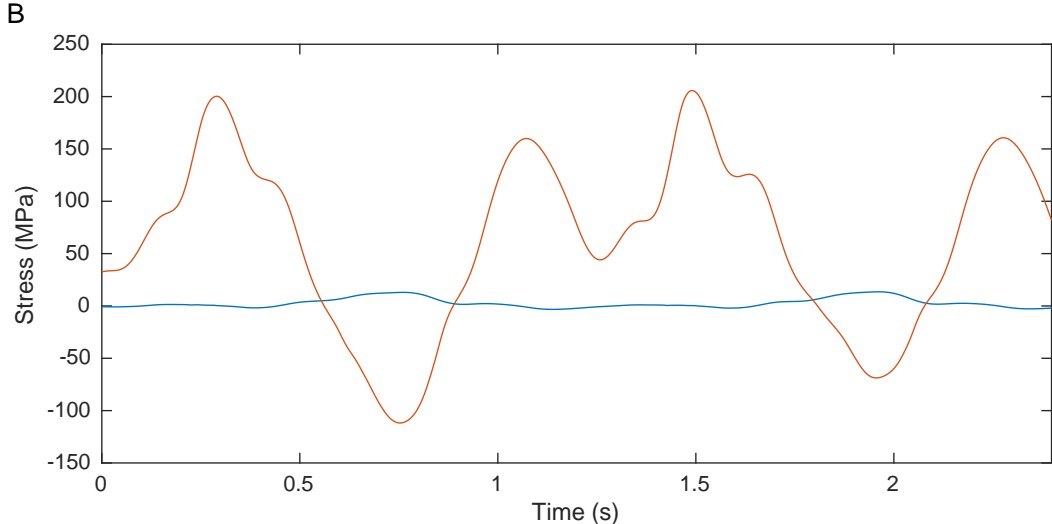

**Figure 1** Graph showing the femoral midshaft stress calculated from the joint reaction forces (2nd order Butterworth low-pass filtered at 5 Hz) from previously published models of (A) *S. camelus* and (B) *T. rex* (*Sellers & Manning, 2007*) using standard equations (*Alexander, 1974*). Ostrich bone cross sections parameters were scaled from the literature (*Brassey et al., 2013b*). Derivation of *T. rex* cross section parameters are explained in the methods section.

basic methods used to construct these models have been described in detail elsewhere (*Sellers et al., 2009*; *Sellers et al., 2013*) but the basic process is outlined below. The model used here was based on a 3D laser scan of BHI 3033 (*Bates et al., 2009*) and consisted of 15 independent segments: a single aggregated trunk segment, along with left and right thigh, shank, metatarsal and pes segments in the hind limb as well as arm, forearm and manus segments in the forelimb. All segments were linked by hinge joints that permitted only pure flexion-extension. This is currently a necessary simplification because our current control system is not able to cope with a fully mobile limbs, but since the main joint

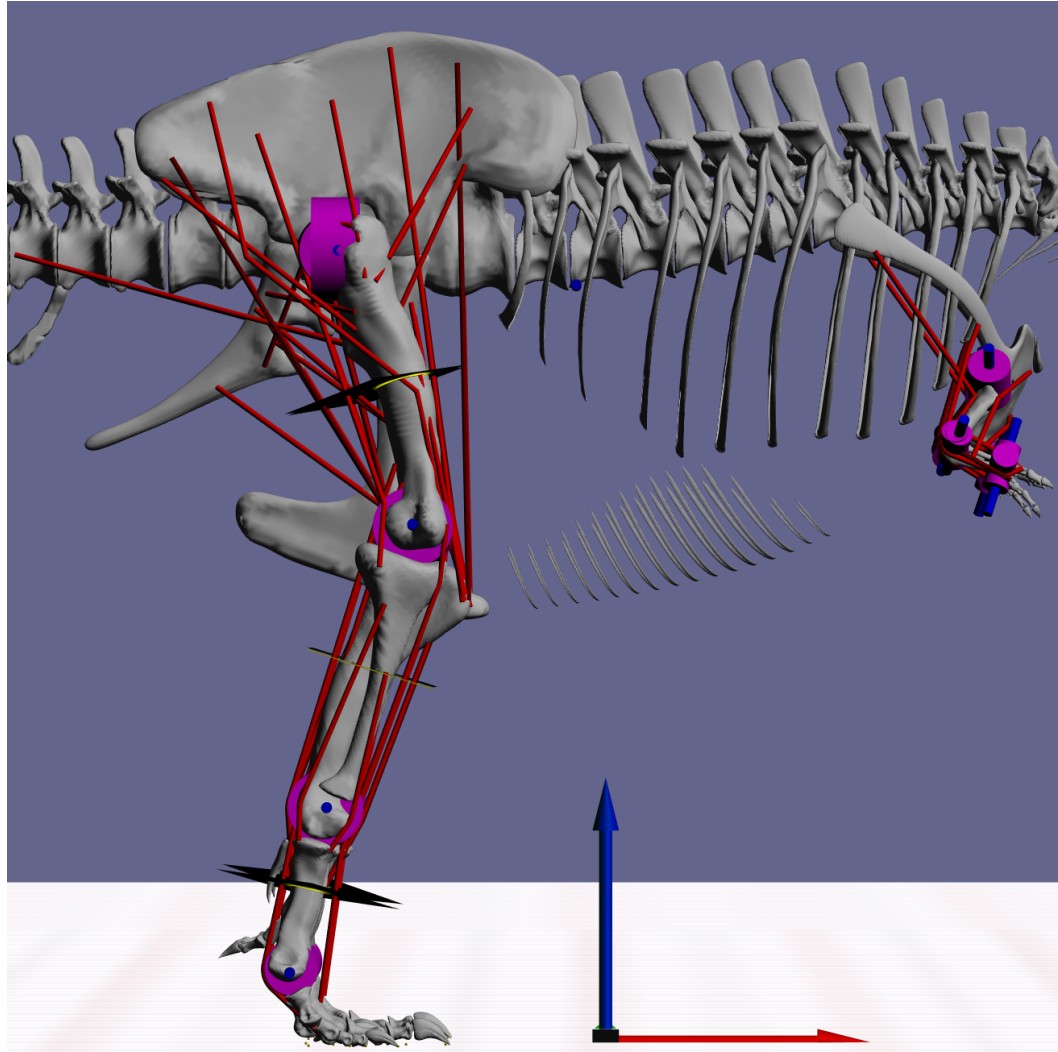

**Figure 2** **Snapshot from GaitSym2016 showing the details of the model.** Muscle paths are in red and joints are in blue. The axes arrows are 1 m long.

actions are likely to be in the parasagittal plane in any case, we do not expect this to affect the predictions to any great extent. Joint positions and ranges of motion were estimated directly from the skeleton. The origins, insertions and paths of 58 hindlimb muscles (29 per limb) were mapped onto the skeletal model based on the comparative analysis of hindlimb muscles in related extant species presented in previous studies (*Hutchinson et al., 2005*). In this simulation a highly simplified forelimb musculature was used since the forelimb was not judged to have an important locomotor role. Muscle mass properties were estimated following the simplified pattern where each muscle action (flexion and extension) and joint location (proximal, intermediate and distal) is considered to have a specific fraction of the total body mass as calculated from a range of extant vertebrates (*Sellers et al., 2013*). The total muscle mass was set at the highest plausible value of 50% (*Sellers & Manning, 2007*) since the current simulation methodology (see below) is only

minimally sensitive to the actual muscle proportion as long as there is sufficient muscle to power the movement. Muscle fibre lengths and tendon lengths were set as a proportion of the change in each muscle–tendon unit length across the range of joint permitted. This setting tunes the actions of the muscles and tendons so that they operate at the most effective parts of their length/tension curves and this minimises the effects of errors in moment arms and lines of action (*Sellers et al., 2009*; *Sellers et al., 2013*). Body mass was estimated from the minimum convex hull of the individual segments using a regression curve calculated from our combined comparative dataset (*Brassey et al., 2013a*; *Brassey et al., 2016*; *Brassey & Sellers, 2014*; *Sellers et al., 2012*), and resulted in a total body mass of 7206.7 kg, which is towards the lower-end of recent estimates from volumetric models (*Bates et al., 2009*; *Hutchinson et al., 2011*). Further information about this calculation and the full calibration dataset is included in the Supplementary Information. Limb segment masses were calculated using the limb mass fractions of total body mass based on running bird data (*Hutchinson, 2004a*). Inertial properties were calculated directly from the convex hulls and scaled to match the predicted masses. The ability to calculate a complete set of inertial properties is one of the major advantages of using volumetric methods for mass estimation in the biomechanical context. Contact with the substrate was modelled using contact spheres attached to the digits as in previous studies (e.g., *Sellers & Manning, 2007*; *Sellers et al., 2009*; *Sellers et al., 2013*). These contacts act like stiff, damped springs under compression, but allow the foot to be lifted with no resistance when needed. However they do not attempt to model the complex, non-linear interactions that actually occur between the foot and the ground.

Bone stress analysis was performed by treating the limb long bones as irregular beams and calculating the mid-shaft loading. The load was calculated directly from the multibody simulator by splitting each of the leg segments into two separate bodies that were linked by a fixed joint. The simulator was then able to calculate both the linear forces and rotational torques acting around this non-mobile joint using the full dynamic model and therefore including inertial forces as well as muscle forces and joint reaction forces. A full finite element analysis would have been preferable but this is currently too computationally expensive in this context and previous work has shown that the mean error in long bone loading is likely to be approximately 10% (*Brassey et al., 2013c*). Bone stress was calculated following Alexander as the sum of the compressive/tensile stress and the normal bending stress (*Alexander, 1974*; *Pilkey, 2002*).

$$\sigma_{\text{compressive}} = \frac{F}{A} \tag{1}$$

where: $\sigma_{\text{compressive}}$ is the normal stress in the beam due to compression (N m$^{-2}$). $F$ is the longitudinal force (N). $A$ is the cross-sectional area of bone (m$^2$).

$$\sigma_{\text{bending}} = \frac{M_x I_y + M_y I_{xy}}{I_x I_y - I_{xy}^2} y - \frac{M_y I_x + M_x I_{xy}}{I_x I_y - I_{xy}^2} x \tag{2}$$

where: $\sigma_{\text{bending}}$ is the normal stress in the beam due to bending (N m$^{-2}$). $x$ is the perpendicular distance to the centroidal $y$-axis (m). $y$ the perpendicular distance to the

centroidal $x$-axis. $M_x$ is the bending moment about the $x$-axis (N m). $M_y$ is the bending moment about the $y$-axis (N m). $I_x$ is the second moment of area about $x$-axis (m$^4$). $I_y$ is the second moment of area about $y$-axis (m$^4$). $I_{xy}$ is the product moment of area (m$^4$).

Equation (1) is the standard equation for calculating compressive stress. Equation (2) calculates the bending stress at a specific $x$, $y$ location in an arbitrary cross-section. The stress calculated by summing the stress values from these two equations ignores the effects of shear but previous work has identified bending and compression as the main loading modes (*Brassey et al., 2013c*). This calculation requires an estimate of the cross-sectional geometry of the limb bones and ideally this would have been obtained directly from a CT scan of the specimen. However since this was not available it was estimated using published cross sectional parameters (or crossectional images if the required measurements were not directly reported) of tyrannosaurs (*Farlow, Smith & Robinson, 1995*; *Horner & Padian, 2004*; *Snively & Russell, 2002b*). This produces a mean cortical thickness for the femur of 38% of the mean external radius, a mean thickness for the tibia of 35%, a mean thickness for the fibula of 96%, and for each bone in the metatarsus 60%. These percentages were converted to actual values using the external outline measured from the laser-scan based reconstruction using custom written python scripts. The complete simulation was implemented in our open source GaitSym system.

To calculate the dynamic loads, this simulation needs to be able to walk and run bipedally. This was achieved using our standard gait morphing methodology (*Sellers et al., 2004*) to generate the necessary control parameters to maximise forward velocity. This process is computationally extremely expensive because of the large number of muscles that are in the model and because of the available degrees of freedom within the model. To reduce the computational difficulty the model was restricted to the parasagittal plane which we have previously found to greatly simplify the control process whilst being unlikely to greatly affect the limb loading (*Sellers et al., 2010*). Even so, finding a stable solution required a great deal of computer time and generating a stable gait took approximately 5,000 core hours before the gait morphing process. The additional constraint of keeping the bone stress below a particular value was implemented by using the peak limb bone stress as a hard fail criteria in the simulator. The stress value was measured across the three major hind limb segments and low-pass filtered at 5 Hz before testing to account for the lack of soft-tissue cushioning in the model and to reflect the level of filtering typically employed in neotological gait analysis (*Winter, 1990*). The simulation was run at a range of different maximum peak stress values using gait morphing to fully investigate the effects of changing this limit on the maximum running speed obtainable. In total over 200 individual optimisation runs were performed to ensure that the search space was adequately covered and that a reasonable estimate of the best performance had been obtained. The full specification of the model is available as a portable, human-readable XML file in the Supplementary Information. The skeletal element outlines and hulls are downloadable from http://www.animalsimulation.org and as Supplementary Information. The model specification is complete and can easily be ported to alternative simulation systems if desired.

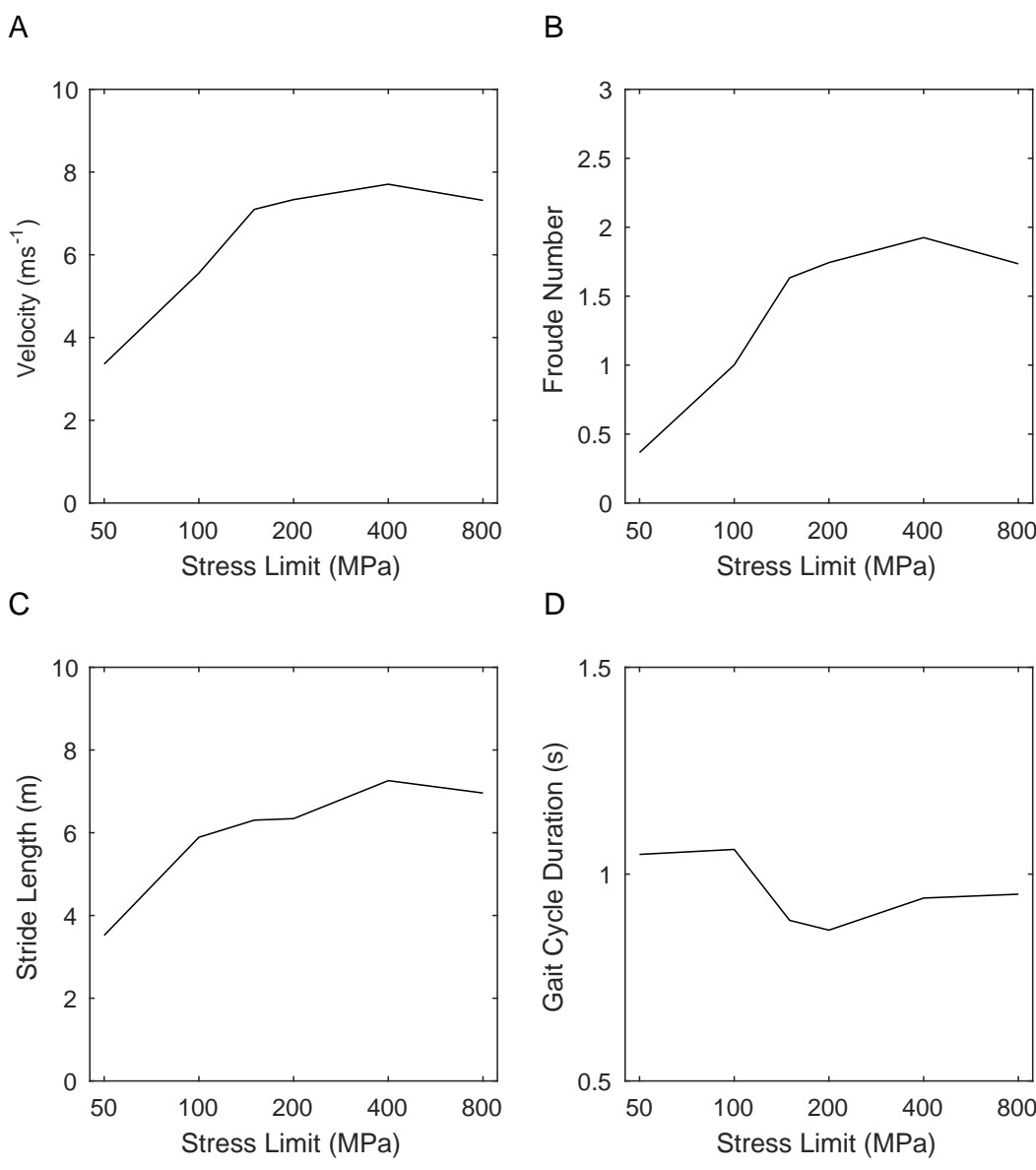

**Figure 3** **Graphs showing the effects of changing the peak stress limit on gait parameters.** (A) maximum velocity; (B) Froude number; (C) stride length; (D) gait cycle duration.

## RESULTS

Figure 3 shows the results of repeated gait morphing whilst optimising for distance travelled in a fixed amount of time using a range of peak stress limits. Figure 3A shows the maximum velocities achieved, which peaks at a speed of 7.7 ms$^{-1}$ for the high stress limit conditions (>200 MPa). Lowering the peak stress limit has little effect on this maximum speed until it is reduced below 150 MPa when the maximum speed drops rapidly. This clearly shows that limiting the stress at high values has no effect on running speed and therefore the simulation is not stress limited in these conditions. At lower stress limits, the stress limit controls the maximum speed indicating that the simulation is stress limited at physiologically realistic

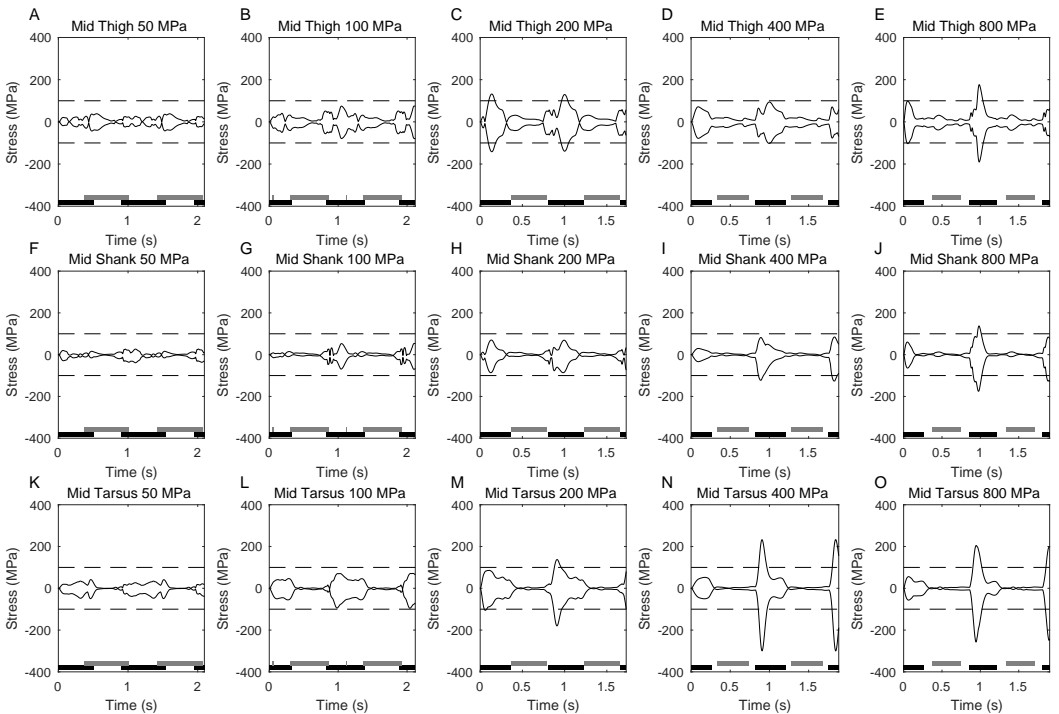

**Figure 4** Graphs showing the peak stress (2nd order Butterworth low-pass filtered at 5 Hz) calculated at the functional mid-point of the hindlimb segments at different peak stress cutoffs. Foot contact times are also shown (black is ipselateral limb, grey is contralateral limb). The time axis represents two complete gait cycles, and the dashed line is drawn at 100 MPa which is the nominal stress limit for a safety factor of 2.

peak stresses. Figure 3B shows the Froude Number calculated from the horizontal velocity and standing hip height. Froude Number in this context is a measure of speed that controls for body size and is therefore useful for cross species comparisons in running velocity (*Biewener & Gatesy, 1991*). From this we can see that the Froude number at 100 MPa is 1.0 which typically is the upper limit for walking gaits. Figure 3C shows the stride lengths adopted by the model. These are broadly in line with Froude number based predictions (*Alexander, 1976*) and show a steady decrease with speed as expected. Figure 3D shows that the gait cycle time is relatively constant in the simulations.

Figure 4 shows the actual peak stresses calculated in the limb during the complete gait cycle as well as showing the periods of foot contact. Relatively high stresses are seen in all the long bones but it is clearly the stress in the mid-tarsus that is highest at high speeds (Figs. 3M–3O). As expected the highest stresses occur during stance phase and the relative symmetry of the maximum and minimum stresses seen at any time show that this stress is primarily due to bending and not to compressive loading on the limb. The foot contact timings confirm the predictions from the Froude numbers that the higher speeds have a clear aerial phase and represent running (i.e., duty factors < 0.5) whereas the slower speeds have no aerial phase and represent a grounded gait (i.e., duty factors > 0.5). The 400 & 800 MPa limit cases are almost identical and the peak stress does not reach 400 MPa again showing that stress is not a limiting factor in these cases.

There are two definitions of walking and running that are commonly used when considering bipedal gait. The traditional definition is "progress by lifting and setting down each foot in turn, so as to have one foot always on the ground" (*Shorter Oxford English Dictionary, 2007*). This definition translates to duty factors: walking has a duty factor of >0.5 and therefore has a period of dual support, whereas running has a duty factor of <0.5 and therefore has an aerial phase (*Alexander, 1984*). However, it is also possible to define bipedal gait based on the energy transformations that are seen between kinetic and gravitational potential energy (*Cavagna, Heglund & Taylor, 1977*). This allows the definition of hybrid gaits such as grounded running where the movements of the centre of mass are typical of running and yet the animal is able to achieve this whilst avoiding an aerial phase in their gait—a gait style commonly seen in birds (*Andrada et al., 2015*). We can perform a similar analysis on our *T. rex* simulation to further investigate the gaits generated. Figure 5 shows the horizontal speed of the centre of mass of the simulation and also the vertical height of the centre of mass. We can measure the phase difference between the kinetic and gravitational potential energy using autocorrelation. At the lowest speed there is a 22% phase difference between these two which drops to <15% at higher speeds. This would indicate moderate energy exchange at low speeds as might be expected. However Fig. 6 shows the actual horizontal kinetic energy of the simulations and the gravitational potential energy and it can be clearly seen that because of the difference in magnitude of the values there is actually very little scope for energy recovery. When constrained by leg stress, the simulation appears to minimise the vertical movement of the centre of mass so that very little gravitational potential energy is ever stored. Our simulation is therefore not taking advantage of pendular energy saving mechanisms which might reflect a preference for grounded running, or it might alternatively be that the model optimisation is for maximum speed and not for minimum energy cost and this has led grounded running to minimise the leg stress as opposed to pendular walking to minimise energy cost.

In the Supplementary Information there are two movie files illustrating the output of the simulator for the fast grounded gait at 100 MPa limit (S2), and the fast run at 400 MPa limit (S3). The full model specification for the models that generated these movies are also available in S1 (S3 and S4).

## DISCUSSION

The velocity changes in Fig. 4 clearly show the marked difference in peak load when comparing walking with running gaits. Extensive work on safety factors in cursorial vertebrates suggests that bone would have a typical maximum stress of not more than 100 MPa (18). In our simulations, fast walking leads to stresses that match this prediction well (Figs. 2 and 3). However all simulations with true running gaits show a large jump in maximum peak stresses that clearly exceeds the maximum allowable value. Body accelerations are higher in running and the force during the contact phase must also be higher because the duty factor is lower. In contrast, accelerations in walking are lower and the increased duty factor reduces forces, and slow walking allows a substantial double support phase so the load on the legs can be divided between both limbs. These factors

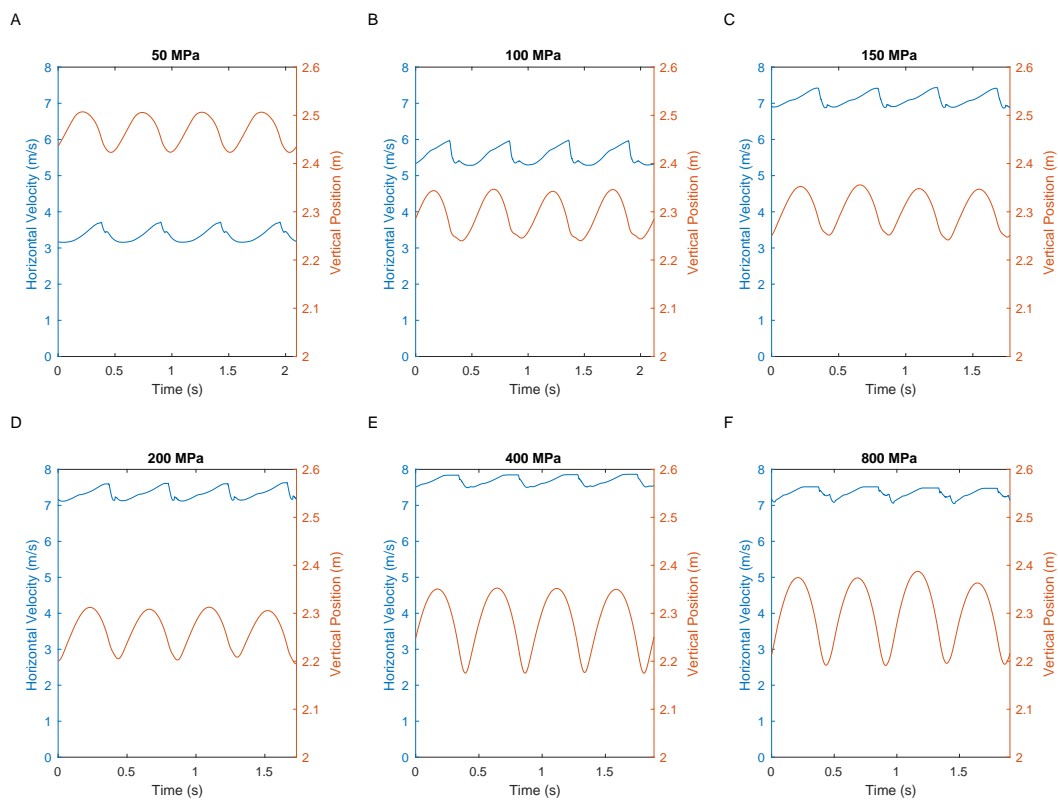

**Figure 5** These graphs show the centre of mass horizontal velocities and the centre of mass vertical positions in the different peak load simulations.

acting together produce the sharp increase noted in peak load in aerial running and, based on the typical stress limits in living animals (*Biewener, 1990*), the skeleton is not strong enough to cope with this load level. Therefore, even if safety factors below the lower limit seen in living animals are allowed, our analysis demonstrates that *T. rex* was not mechanically capable of true running gaits (Figs. 2 and 3). Previous quantitative estimates of absolute maximal speeds for *T. rex* ranged from 5-15m/s (*Gatesy, Baker & Hutchinson, 2009*; *Hutchinson, 2004b*; *Hutchinson & Garcia, 2002*; *Sellers & Manning, 2007*) and identified soft tissue unknowns as a major source of uncertainty but by including hard tissue mechanical information we can show that the highest values, whilst possible if we allow generous estimates for soft tissue, are impossible given skeletal strength. Bone strength is based directly on the skeletal dimensions and in our analysis of *T. rex* the forces generated by the muscles are not limiting the top speed. When extremely high stresses are permitted in the model (>150 MPa, and especially 400–800 MPa) then predicted speeds are consistent with mean estimates from previous models in which only muscular constraint on maximal performance are considered (*Sellers & Manning, 2007*). In addition our analysis of energy transformations (Figs. 4 and 5) further reinforces the suggestion that the simulation is finding solutions that minimise the skeletal load and that low impact, bird-style grounded running (*Andrada et al., 2015*) may be an appropriate gait for bipedal dinosaurs.

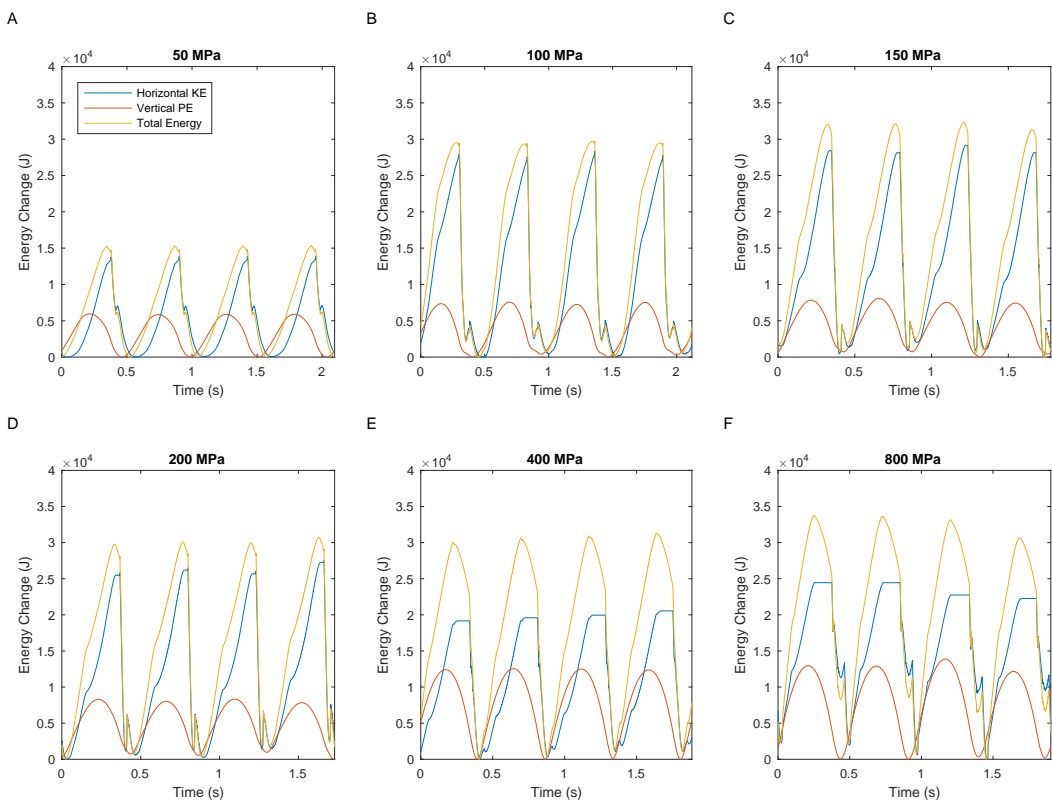

**Figure 6** These graphs show the energy transformations within the simulation: horizontal kinetic energy, gravitational potential energy, and also the sum of these two energy values.

As with all attempts at reconstructing the locomotor capabilities of fossil animals it is important to be somewhat cautious with our interpretations. These results improve on those obtained by previous biomechanical work by excluding some of the previously plausible values and thereby reducing the range of uncertainty but many of the previous caveats still apply. Our previous work on sensitivity analysis (*Bates et al., 2010*) tested the effects of body mass, centre of mass location and various measures of muscle physiology, but these complex models have a large number of additional parameters that could potentially affect the model predictions. Ideally a full Monte Carlo style sensitivity analysis would be performed to analyse the effects of all of these parameters (*Campolongo et al., 2000*) but unfortunately the computational requirement for such an analysis is enormous and currently not a practical undertaking. It would also be useful to ground truth our predictions based on experimental work with living animals. Direct bone strain measurement is a well-established technique that has been performed on a wide range of animals (e.g., *Biewener & Taylor, 1986*; *Burr et al., 1996*; *Main & Biewener, 2006*; *Rubin & Lanyon, 1982*) and multibody dynamic analysis derived strains have been validated against the literature in several cases (e.g., *Al Nazer et al., 2008*; *Curtis et al., 2008*) but there is certainly a need to combine these approaches in the same experimental system and this would be a useful future approach. The models used in our simulations are currently the most anatomically complete reconstructions ever attempted. However, they are still appreciable simplifications of the true complexity

of the living organism. In particular, extending the stress analysis to a full finite element model would be of considerable benefit especially if coupled with a more realistic muscle coverage achieved by subdividing anatomical muscles into multiple functional units and by including other non-bone tissues. These extra elements would potentially allow the model to exploit the possibilities of peak stress reduction using soft tissue tensile elements to produce tensegrity structures (*Schilder, 2016*) which might turn out to have a substantial effect as has been suggested in other tyrannosaurids (*Snively & Russell, 2002a*). Finally, our simulations rely on machine learning to find muscle activation patterns that maximise the speed given a range of constraints. There are too many possible combinations of parameters to perform an exhaustive search over all possibilities so we need to use a non-exhaustive approach. This is a very active area of current computational research and we would certainly expect that better solutions will be found using a combination of the improved algorithms and the greater computational power which will be available in future.

The finding that *T. rex* was restricted to walking gaits supports arguments for a less athletic lifestyle for the largest bipedal dinosaurs like *T. rex. Tyrannosaurs* underwent pronounced allometric changes during ontogeny (*Brusatte et al., 2010*) and previous studies have suggested the torso became longer and heavier whereas the limbs became proportionately shorter and lighter as *T. rex* grew (*Hutchinson et al., 2011*). It would therefore be very valuable not only to investigate other species but also apply our multiphysics approach to different growth stages within species. Ontogenetic niche partitioning has been suggested for many dinosaurs (*Fricke & Pearson, 2008*; *Lyson & Longrich, 2011*), and energetic considerations (*Horner, Goodwin & Myhrvold, 2011*) and changes in skull anatomy (*Carr, 1999*) and bite performance (*Bates & Falkingham, 2012*) may indicate a shift towards increased consumption of larger prey and/or carrion as *T. rex* grew. Such a shift towards large prey specialism is not incompatible with our findings here regarding locomotor speed, as presumably large multi-tonne herbivores similarly experienced the same general scaling-related restrictions on musculoskeletal performance as *T. rex* (*Bates, Benson & Falkingham, 2012*; *Bates et al., 2010*; *Hutchinson, 2004b*; *Hutchinson et al., 2005*; *Hutchinson & Garcia, 2002*; *Sellers & Manning, 2007*). There certainly appears to be direct evidence of predatory behaviour in *T. rex* (*Carpenter, 1998*; J 2008) which supports the idea of predator prey interactions concerning locomotor performance. It is somewhat paradoxical that the relatively long and gracile limbs of *T. rex*—long argued to indicate competent running ability (*Bakker, 1986*; *Christiansen, 1998*; *Paul, 1998*; *Paul, 2008*)—would actually have mechanically limited it to walking gaits, and indeed maximised its walking speed. This observation illustrates the limitation of approaches that rely solely on analogy and the importance of a full biomechanical analysis when investigating animals with extreme morphologies such as *T. rex*. The new approach we introduce here clearly has the potential to contribute widely to our understanding of the evolution of animal locomotion, particularly major ecological shifts such as colonization of land or bipedal-quadrupedal transitions.

## CONCLUSION

The results presented demonstrate that the range of speeds predicted by earlier biomechanical models for *T. rex* locomotion include speeds that would apply greater loads to the skeleton than it would have been able to withstand. These high load speeds can therefore be excluded from our predictions and this means that the possible range of maximum speeds has been greatly reduced and essentially limits adults of this species to walking gaits. This finding may well generalise to other long-limbed giants such as *Giganotosaurus*, *Mapusaurus*, and *Acrocanthosaurus* but this idea should be tested alongside experimental validation work on extant bipedal species. This work demonstrates how including multiple physical modalities and multiple goals can improve our reconstructions of the locomotor biology of ancient organisms and lead to a better understanding of the mechanical constraints of large body size.

## ACKNOWLEDGEMENTS

The authors would like to thank NERC, the Leverhulme Trust, BBSRC, EPSRC and PRACE for their ongoing support developing GaitSym; the staff at the high performance computer centres where the simulations were carried out (Archer, Hartree and N8); and the staff at the Black Hills Institute for allowing us to scan BHI 3033.

### Funding

Software development for this project was funded by BBSRC (BB/K006029/1), Leverhulme Trust F/00 025/AK, NERC NE/C520447/1. This work made use of the facilities of N8 HPC Centre of Excellence, provided and funded by the N8 consortium and EPSRC (Grant No. EP/K000225/1). The Centre is co-ordinated by the Universities of Leeds and Manchester. The funders had no role in study design, data collection and analysis, decision to publish, or preparation of the manuscript.

### Grant Disclosures

The following grant information was disclosed by the authors:
BBSRC: BB/K006029/1.
Leverhulme Trust: F/00 025/AK.
NERC: NE/C520447/1.
N8 consortium.
EPSRC: EP/K000225/1.

### Competing Interests

The authors declare there are no competing interests.

### Author Contributions

- William I. Sellers conceived and designed the experiments, performed the experiments, analyzed the data, contributed reagents/materials/analysis tools, wrote the paper, prepared figures and/or tables, reviewed drafts of the paper, wrote the software.

- Stuart B. Pond conceived and designed the experiments, contributed reagents/materials/analysis tools, wrote the paper, prepared figures and/or tables, reviewed drafts of the paper, created the 3D skeleton.
- Charlotte A. Brassey, Philip L. Manning and Karl T. Bates conceived and designed the experiments, contributed reagents/materials/analysis tools, wrote the paper, reviewed drafts of the paper.

## Data Availability

Figshare: https://figshare.com/articles/Investigating_the_running_abilities_of_Tyrannosaurus_rex_using_stress-constrained_multibody_dynamic_analysis/4700275.

## Supplemental Information

Supplemental information for this article can be found online at http://dx.doi.org/10.7717/peerj.3420#supplemental-information.

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
