# Peer review of "Investigating the running abilities of Tyrannosaurus rex using stress-constrained multibody dynamic analysis"

_PeerJ, doi:10.7717/peerj.3420_

## Round 0.1 · original submission · Minor Revisions

Congratulations - both reviewers found much of value in your manuscript. They also made many suggestions for improvement, all of which strike me as reasonable. Please be attentive in addressing the points they raised.

In particular, Reviewer 2 is correct that some kind of validation study on an extant model will be required. He proposed three options, and I strongly encourage you to pursue all three (especially since the first and third are simply citations of existing literature). This is not a case of a biologist being "terribly skeptical about simulations with musculoskeletal parameters", but rather of an editor who wants to see work published which has survived at least one test in the real world.

As hinted by Reviewer 2, the manuscript is concise to the point of being a difficult read in some places. I assume, based on the numbered references (which are not used at PeerJ), that the manuscript was previously submitted to a higher rung on the publication ladder. That's fine and even flattering, if my suspicion about the identity of that outlet is correct. And it opens up a couple of opportunities: first, to relax the tightness of the prose into something a bit more readable, now that you're not under a punitive word-length requirement, and second, to make sure that the revised manuscript conforms to PeerJ format, particularly the in-text citations and References section.

Normally in a situation like this, I'd return a "major revisions" decision, especially given the requirement - which I regard as mandatory - to include a validation study. After much consideration, and with some reluctance, I am following the recommendation of the reviewers and flagging this for "minor revisions". Whether it requires another round of review will depend on your diligence in addressing the concerns noted above. Please take all of these comments in the constructive spirit in which they are intended. I will look forward to seeing an improved version of this work in the future.

·

Basic reporting

This is a very interesting attempt to model the running ability--or lack thereof--of the huge theropod dinosaur _Tyrannosaurus rex_. I am not qualified to evaluate the details of the models, but as best I can tell they are reasonable, and generate plausible results.

The only problems with the manuscript that I see are minor, typographic errors and the like:

1) Line 90: should probably read "the maximum" rather than "maximum the"

2) LIne 104: change "simulations" to "simulation"

3) Line 222: seems to be a missing word after "measured"

4) Line 227: change "minimises" to "minimise"

5) Line 241: Should refer to Figure 3 rather than Figure 1

6) Line 262: Is the word "results" supposed to be there?

7) Line 290: change "and" to "an"

8) Line 311: change "on solely" to "solely on"

Experimental design

As best I can tell this contribution meets all of your standards. I would suggest, however, that--if you have not already done so--you have the model evaluated by someone with a greater knowledge of this kind of modeling than myself.

Validity of the findings

I see no problems here.

Additional comments

A very interesting contribution to an equally interesting problem in paleobiology

·

Basic reporting

The manuscript meets nearly all criteria quite well. Some additional references are suggested in the review pdf.

The discussion of grounded running is incongruous in the results section. Distribute and modify some of this text appropriately to other sections (including intro and discussion).

Experimental design

The methods are sparse, but careful reading shows that they're concise rather than incomplete (although see suggestions below).

Why sum compressive stress and normal bending stress? Bending will cause compression and tension on either side of the neutral surface/section. Does your equation (2) for bending stress ignore or incorporate compressive stress? Double check these equations and your approach, but please do not assume that I am omniscient and that your approach is wrong. (That’s the path to unnecessary anxiety and delay.) It's been a long time since my structural mechanics courses, in which I learned to calculate stress at any point in any structure given its shape and applied moments. Your way may work just as well.

Validity of the findings

The results are sound. However, you will have to deal with complaints about the lack of validation. Biologists are terribly skeptical about simulations with musculoskeletal parameters, even when you’re carefully and explicitly optimizing within tremendous ranges for most necessary criteria.

Here are several options for addressing such objections, one or more of which you can try.
1. Note that similar methods are validated for ostriches (to keep it within Dinosauria), citing Rankin et al. (2016) and their replication of actual ostrich locomotion with inverse dynamics to optimize forward dynamics. That may be enough, although they did not restrict motion to parasagittal movement of the limbs.
2. Calculate stresses similarly in at least one bone from your earlier ostrich model (and maybe human model), to show that smaller critters may not have the same stress constraints as do giant theropods. Gilbert et al. (2016) found good safety factors in an ostrich tarsometatarsus, using FEA. I am happy to send ostrich CT scans if you want to calculate structural properties, and use those and reaction forces to estimate stresses. (Scaling to the size of your ostrich, of course).
3. Be careful to cite Brassey et al. (2013)’s comparison of FEA and beam theory, to show that you’re thoroughly cognizant of the range of error.

Options 2 and 3 may satisfy the calls for validation of stresses. Obvious to us, but unfamiliar to reviewers who are not specialists. I’ve found it worthwhile to take the opportunity to educate, rather than ignore or dismiss. If you’re writing a response to reviews for the editor, you’re likely explaining all this anyway.

Your conclusion can be stronger if you invoke other giant carnivorous dinosaurs, whose leg bones are more slender than those of T. rex of comparable weight. I don't have numbers, but see skeletal reconstructions by Hartman and Paul. Snively, Russell, and Powell (2004) show a Mapusaurus (pre-naming) third metatarsal that is distally more slender than a T. rex MT III, particularly anteroposteriorly.

Additional comments

The paper's combination of dynamics and structural mechanics is enlightening about constraints on running (and favorable walking gaits) in large bipedal dinosaurs. My major advice is under "Validity of the findings" above. Along with mostly minor (non-essential) suggestions for diction, the manuscript will improve with the following adjustments.

1. On page 2, you might cite more papers on locomotor FEA, including an early one on tyrannosaurs. The text and citations are fine as they are; modify at your discretion.

2. Page 2, line 54. What do you mean by mass fractions for limb segments? Is this as a percentage of body mass?

3. Page 3, lines 44-46. What do the percentages mean in your cross-sectional parameters? The femur mean is 38% of what? The metatarsal cross-sections (reference 41) were areas of intermetatarsal ligament attachments. Go with your data from external dimensions.

4. Page 4, lines 44-52. This paragraph starts with a discussion of gaits, which is more appropriate to the Introduction.

5. Page 6, lines 1-3. The (arcto-)metatarsus of tyrannosaurids had extensive ligaments that would transduce bending loads on the metatarsals into tension on these structures, for a tensegrity-like effect. The metatarsus seems to be a weak point, but complex stuff was going on in there. The text has a brief suggested addition that addresses these points (you cite the pertinent reference elsewhere in the manuscript.

---

## Round 0.2 · accepted · Accept

Thank you for your diligence in addressing the concerns of the reviewers. I am satisfied with the revised manuscript, and I am happy to accept it for publication in PeerJ.

The decision of whether or not to publish the peer reviews alongside the paper is entirely yours, and will not affect how your paper is handled going forward. However, I encourage you to do so. It's a great example of a constructive review process. More importantly, both reviewers chose to sign their reviews, and making the reviews public allows the reviewers to receive more credit for their efforts. It also contributes to the emerging culture of fairness and transparency in editing and peer review.